# LoRA-GA: Low-Rank Adaptation with Gradient Approximation

**Shaowen Wang**
wangsw23@mails.tsinghua.edu.cn

**Linxi Yu**
yulx23@mails.tsinghua.edu.cn

**Jian Li** *
lijian83@mail.tsinghua.edu.cn
Tsinghua University
Beijing, China

## Abstract

Fine-tuning large-scale pretrained models is prohibitively expensive in terms of computational and memory costs. LoRA, as one of the most popular Parameter-Efficient Fine-Tuning (PEFT) methods, offers a cost-effective alternative by fine-tuning an auxiliary low-rank model that has significantly fewer parameters. Although LoRA reduces the computational and memory requirements significantly at each iteration, extensive empirical evidence indicates that it converges at a considerably slower rate compared to full fine-tuning, ultimately leading to increased overall compute and often worse test performance. In our paper, we perform an in-depth investigation of the initialization method of LoRA and show that careful initialization (without any change of the architecture and the training algorithm) can significantly enhance both efficiency and performance. In particular, we introduce a novel initialization method, LoRA-GA (**Low R**ank **A**daptation with **G**radient **A**pproximation), which aligns the gradients of low-rank matrix product with those of full fine-tuning at the first step. Our extensive experiments demonstrate that LoRA-GA achieves a convergence rate comparable to that of full fine-tuning (hence being significantly faster than vanilla LoRA as well as various recent improvements) while simultaneously attaining comparable or even better performance. For example, on the subset of the GLUE dataset with T5-Base, LoRA-GA outperforms LoRA by 5.69% on average. On larger models such as Llama 2-7B, LoRA-GA shows performance improvements of 0.34, 11.52%, and 5.05% on MT-bench, GSM8K, and Human-eval, respectively. Additionally, we observe up to 2-4 times convergence speed improvement compared to vanilla LoRA, validating its effectiveness in accelerating convergence and enhancing model performance. Code is available at code.

## 1 Introduction

Fine-tuning large language models (LLMs) is essential for enabling advanced techniques such as instruction fine-tuning [1], reinforcement learning from human feedback (RLHF) [2], and adapting models to specific downstream applications. However, the computational and storage costs associated with full fine-tuning are prohibitively high, particularly as model sizes continue to grow. To address these challenges, methods of Parameter-Efficient Fine-Tuning (PEFT) (see e.g., [3]), such as Low-Rank Adaptation (LoRA) [4], have emerged and gained significant attention.

---

*Corresponding author

38th Conference on Neural Information Processing Systems (NeurIPS 2024).

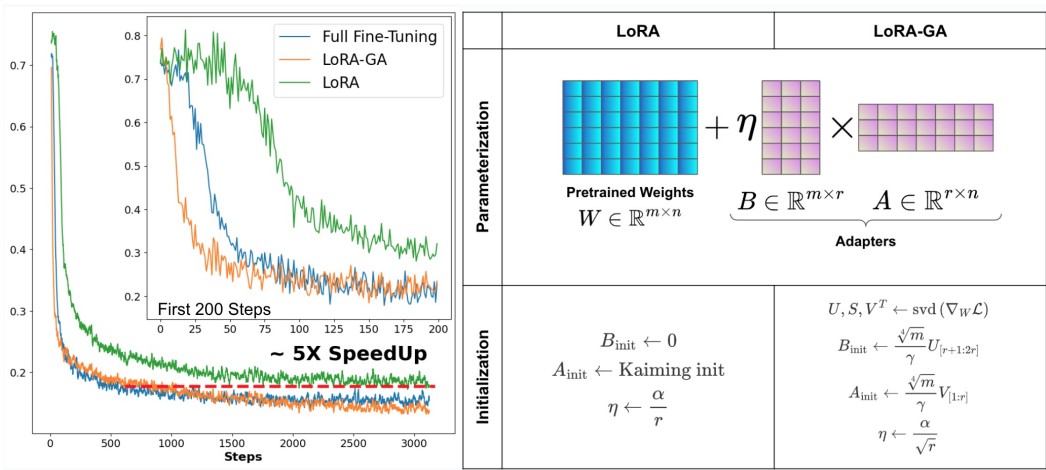

Figure 1: (**Left**) Training loss curves of Llama 2-7B on MetaMathQA to training steps. LoRA-GA converges as quickly as full fine-tuning and outperforms LoRA. (**Right**) Initialization procedures used in LoRA and LoRA-GA. The key difference is that LoRA-GA initializes adapters using the eigenvectors of the gradient matrix, as opposed to random initialization with a scaling factor.

Instead of updating the parameters of the model directly, LoRA incorporates auxilary low-rank matrices $B$ and $A$ into the linear layers of models (such as the $Q, K, V$, and $O$ matrices in a self-attention block [5]), while keeping the original layer weights $W$ fixed. The modified layer is represented as $y = (W + \eta BA)x$, where $x$ is the input of that layer, $y$ is the output, and $\eta$ is the scaling factor. This approach significantly reduces the number of parameters that need to be fine-tuned, thereby lowering the computational and memory costs at each step.

Despite these benefits, extensive empirical evidence (see e.g., [6, 7, 8, 9]) shows that LoRA converges significantly slower compared to full finetune. This slower convergence often increases overall computational costs (measured in Floating Point Operations) and can sometimes lead to worse test performance. In our experiments, we typically observe that LoRA requires 5-6x more iterations and FLOPs to reach the same performance as full fine-tuning under the same learning rate, as shown in Figure 1.

To study the cause of slow convergence, we perform an in-depth investigation of the initialization strategy of LoRA's adapter weights. It is known that fine-tuning pretrained models using the same objective (e.g., language modeling) often converges faster than re-initializing new parameters (e.g., a classification head) [10]. This observation leads us to question whether the slow convergence of vanilla LoRA might be attributed to the default random initialization of adapter weights (LoRA initializes $A$ using Kaiming initialization [11] and sets $B$ to zero [4]). In our experiments, we find that different initialization strategies for LoRA can significantly impact the results, and its default initialization is suboptimal.

In pursuit of a convergence rate comparable to full fine-tuning, we aim for initialization so that the update of $BA$ matches the update of $W$ closely. Previous work suggests that gradient descent operates in a low-dimensional subspace [12, 13]. If we can closely approximate the gradients of the full model at the initial step, subsequent steps can also be approximated, potentially accelerating the convergence of LoRA.

To this end, we introduce a novel initialization method, LoRA-GA (**Lo**w **Ra**nk **G**radient **A**pproximation). By initializing $A_{\text{init}}$ and $B_{\text{init}}$ with the eigenvectors of the full gradient matrix, the gradient of the low-rank product $BA$ aligns with the direction of the gradient of the full weight matrix $W$. Mathematically, we aim to ensure that:

$$\Delta(BA) \approx \zeta \Delta W, \quad \text{for some non-zero positive constant } \zeta.$$

**Our contributions can be summarized as follows:**

**1.** We propose LoRA-GA , a novel initialization method for LoRA that accelerates convergence by approximating the gradients of the low-rank matrices with ones of the full weight matrix.

**2.** We identify the scaling factor under non-zero initialization, which ensures the variance of adapter outputs is invariant to the rank of the adapter and the dimension of the input.

**3.** We validate LoRA-GA through extensive experiments, demonstrating significant performance improvements and faster convergence compared to vanilla LoRA. Specifically, LoRA-GA outperforms LoRA by 5.69% on the GLUE [14] subset with T5-Base [15], and by 0.34, 11.52%, and 5.05% on MT-bench [16], GSM8K [17], and HumanEval [18] with Llama 2-7B [19], respectively, while achieving up to 2-4 times faster convergence.

## 2 Related Work

### 2.1 Initialization

The significance of maintaining variance stability during initialization has been widely acknowledged to prevent the occurrence of diminishing or exploding phenomena. Xavier initialization [20] ensures stability in both the forward and backward passes of a network under a linear activation function. He initialization [11] extends this solution to networks using ReLU activation. Distinct from these, LSUV initialization [21] selects a mini-batch of data, performing a forward pass to determine the output variance, and subsequently normalizing it to ensure stability. Tensor program (see e.g., [22]) has emerged as a powerful technique for tuning various hyperparameters, including the initialization, for large models.

### 2.2 Parameter-Efficient Fine-Tuning (PEFT)

To fine-tune increasingly large language models within limited hardware resources, researchers have developed various Parameter-Efficient Fine-Tuning (PEFT) methods. Adapter-based methods [23, 24, 25, 26] incorporate new layers into existing model layers. While fine-tuning only these inserted layers significantly reduces resource consumption and requires much fewer parameters, this approach introduces additional latency during both forward and backward passes. Soft Prompt-based methods [10, 27, 28, 29, 30] prepend learnable soft tokens to the model's input to adapt the model to specific tasks. This approach effectively leverages the pre-trained model's capabilities, requiring only appropriate prompts for task adaptation, though it incurs computational overhead during inference. More broadly, GaLore [31] applies low-rank gradients to parameter updates for memory efficiency during training. While this approach is highly expressive and performant, it requires storing complete model checkpoints, consuming more storage than other PEFT methods.

### 2.3 LoRA's Variants

LoRA is one of the most popular PEFT methods that introduces the product of low-rank matrices alongside existing layers to approximate weight changes during fine-tuning. Several methods have been proposed to improve the structure of LoRA. AdaLoRA [32] dynamically prunes insignificant weights during fine-tuning using SVD, allowing more rank allocation to important areas within a fixed parameter budget. DoRA [8] enhances the model's expressiveness by adding learnable magnitudes to the direction adjustments made by low-rank matrix products. Additionally, LoHA [33] and LoKr [34] employ Hamiltonian and Kronecker products, respectively.

Despite these advancements, vanilla LoRA remains the most popular method due to its robust library and hardware support. Therefore, improving LoRA without altering its structure and at a low cost is crucial. Several recent methods focus on this aspect. ReLoRA [35] suggests periodically merging learned adapters into the weight matrices to enhance LoRA's expressibility. LoRA+ [36] proposes using different learning rates for the two matrices in LoRA to improve convergence. rsLoRA [37] introduces a new scaling factor to make the scale of the output invariant to rank. Although our stable scale approach appears similar to rsLoRA, rsLoRA assumes $BA = 0$ initialization, making $r$ invariant to the update $\Delta BA$. In contrast, our stable scale ensures that non-zero initialized $BA$ remains invariant to both rank and input dimension from the start.

Recently, PiSSA [38] proposes to initializing $A$ and $B$ to approximate the original matrix $W$, by performing SVD on $W$. Our method, however, is based on a very different idea, that is to approximate the gradient of $W$, which involves performing SVD on sampled gradients and properly scaling the initialized matrices, as detailed in Section E.

## 3 Methods

In this section, we analyze the initialization of LoRA and introduce our method, LoRA-GA. LoRA-GA consists of two key components: (i) approximating the direction of the gradient of full finetune and (ii) ensuring rank and scale stability in the initialization process. We examine each component and subsequently present their integration within LoRA-GA.

### 3.1 Review of Vanilla LoRA

**Structure of LoRA**    Based on the hypothesis that the updates of fine-tuning are low-rank [13], LoRA [4] proposes to use the product of two low-rank matrices to represent the incremental part of the original matrix $W$. Here, $W$ is the weight matrix of a linear layer in the model. For example, in transformers, it could be the $Q, K, V,$ or $O$ matrices of the self-attention layer or the weight matrix in the MLP layer. Specifically, LoRA has the following mathematical form:

$$W' = W_0 + \Delta W = W_0 + \frac{\alpha}{r} BA := W_0 + \eta BA$$

where $W', W_0 \in \mathbb{R}^{m \times n}$, $B \in \mathbb{R}^{m \times r}$, and $A \in \mathbb{R}^{r \times n}$, with $r \ll \min(m, n)$. $W_0$ is the pre-trained weight matrix, remains frozen during the fine-tuning process, while $A$ and $B$ are trainable.

**Initialization of LoRA**    Under LoRA's default initialization scheme [4, 39], matrix $A$ is initialized using Kaiming uniform [11], while matrix $B$ is initialized with all zeros. Consequently, $BA = 0$ and $W'_0 = W_0$, ensuring that the initial parameters are unchanged.

If the additional term $\Delta W = \eta BA$ is initially non-zero (e.g., [38]), the frozen parameter can be adjusted to ensure the initial parameters unchanged. This can be expressed as:

$$W' = (W_0 - \eta B_{\text{init}} A_{\text{init}}) + \eta BA := W_{\text{frozen}} + \eta BA$$

where $W_{\text{frozen}} = W_0 - \eta B_{\text{init}} A_{\text{init}}$ is frozen, and $B$ and $A$ are trainable in this case.

### 3.2 Gradient Approximation

Our goal is to ensure that the first-step update $\Delta(\eta BA)$ approximate the direction of the weight update $\Delta W$, i.e., $\Delta(\eta BA) \approx \zeta \Delta W$ for some non-zero positive constant $\zeta$. We will discuss how to choose $\zeta$ in Section 3.3 and one can treat $\zeta$ as a fixed constant for now.

Consider a gradient descent step with learning rate $\lambda$, the updates for $A$ and $B$ are $\Delta A = \lambda \nabla_A \mathcal{L}(A_{\text{init}})$ and $\Delta B = \lambda \nabla_B \mathcal{L}(B_{\text{init}})$, respectively. Assuming learning rate $\lambda$ is small, the update of $\eta BA$ at the first step can be expressed as:

$$\eta(\Delta BA_{\text{init}} + B_{\text{init}} \Delta A) = \eta \lambda [\nabla_B \mathcal{L}(B_{\text{init}}) A_{\text{init}} + B_{\text{init}} \nabla_A \mathcal{L}(A_{\text{init}})]$$

To measure its approximation quality of scaled the update of the weights in full finetune $\zeta \Delta W = \zeta \lambda \nabla_W \mathcal{L}(W_0)$, we use the Frobenius norm of the difference between these two updates as a criterion:

$$
\begin{aligned}
&\|\eta(\Delta BA_{\text{init}} + B_{\text{init}} \Delta A) - \zeta \lambda \nabla_W \mathcal{L}(W_0)\|_F \\
=&\lambda \|\eta \nabla_B \mathcal{L}(B_{\text{init}}) A_{\text{init}} + \eta B_{\text{init}} \nabla_A \mathcal{L}(A_{\text{init}}) - \zeta \nabla_W \mathcal{L}(W_0)\|_F
\end{aligned}
\tag{1}
$$

**Lemma 3.1.** *Suppose the loss function is $\mathcal{L}$ and $y = W'x = (W_0 + \eta BA)x$, where $y$ is the output of a layer and $x$ is the input, the gradients of $A$ and $B$ are linear mappings of the gradient of $W'$:*

$$\nabla_A \mathcal{L} = B^T \nabla_{W'} \mathcal{L}, \quad \nabla_B \mathcal{L} = (\nabla_{W'} \mathcal{L}) A^T$$

*Remarkably, $\nabla_{W'} \mathcal{L}$ in LoRA and $\nabla_W \mathcal{L}$ in full fine-tuning are equal at the beginning of the training.*

By substituting the gradients in Lemma 3.1 into Equation 1, we can rewrite the criterion as follows:

$$\lambda \|\eta^2 \nabla_{W'} \mathcal{L}(W_0) \cdot A_{\text{init}}^T A_{\text{init}} + \eta^2 B_{\text{init}} B_{\text{init}}^T \cdot \nabla_W \mathcal{L}(W_0) - \zeta \nabla_W \mathcal{L}(W_0)\|_F \tag{2}$$

This criterion evaluates how well the adapter's gradient approximates the direction of the gradient of full fine-tuning, and minimizing it brings the gradient of LoRA closer to that of full fine-tuning with a scaling factor $\zeta$:

$$\min_{A_{\text{init}}, B_{\text{init}}} \|\eta^2 \nabla_W \mathcal{L} \cdot A_{\text{init}}^T A_{\text{init}} + \eta^2 B_{\text{init}} B_{\text{init}}^T \cdot \nabla_W \mathcal{L} - \zeta \nabla_W \mathcal{L}\|_F \tag{3}$$

**Theorem 3.1.** *For the optimization problem in Equation 3 with given $\zeta$, if the Singular Value Decomposition (SVD) of $\nabla_W \mathcal{L}$ is $\nabla_W \mathcal{L} = USV^T$, the solution is:*

$$B_{\text{init}} = \frac{\sqrt{\zeta}}{\eta} U_{I_A}, \quad A_{\text{init}} = \frac{\sqrt{\zeta}}{\eta} V_{I_B}^T, \text{ such that } |I_A| = |I_B| = r, \ I_A \cup I_B = \{i \mid 1 \leq i \leq 2r, i \in \mathbb{N}\}$$

*where $I_A$ and $I_B$ are index sets.*

Theorem 3.1 provides an appropriate initialization scheme for $A_{\text{init}}$ and $B_{\text{init}}$ given a specific $\zeta$. The selection of $\zeta$, which influences the scaling of the update $\eta BA$, will be discussed in the following section.

## 3.3 Scale Stability

Inspired by rsLoRA citekalajdzievski2023rank and the Kaiming initialization [11], we define the following notions of stability:

**Definition 3.1.** *When $d_{out}, d_{in}, r \to \infty$, an adapter $\eta BA$ exhibits two distinct types of scale stabilities:*

*1. **Forward stability**: If the inputs to the adapter are independently and identically distributed (i.i.d.) with 2nd moment $\Theta_{r,d_{out},d_{in}}(1)$, then the 2nd moment of the outputs remains $\Theta_{r,d_{out},d_{in}}(1)$.*

*2. **Backward stability**: If the gradient of the loss with respect to the adapter outputs is $\Theta_{r,d_{out},d_{in}}(1)$, then the gradient with respect to the inputs remains $\Theta_{r,d_{out},d_{in}}(1)$.*

**Theorem 3.2.** *Given the initialization proposed in Theorem 3.1, assume that the orthogonal vectors in $A_{\text{init}}$ and $B_{\text{init}}$ are randomly selected from the unit spheres in $\mathbb{R}^{d_{in}}$ and $\mathbb{R}^{d_{out}}$ with the constraint that the vectors are orthogonal to each other, and $\eta = \Theta_{r,d_{out},d_{in}}(1/\sqrt{r})$ as suggested by rsLoRA [37]. Under these conditions, the adapters are forward scale-stable if $\zeta = \Theta_{r,d_{out},d_{in}}\left(\sqrt{d_{out}/r^2}\right)$ and backward scale-stable if $\zeta = \Theta_{r,d_{out},d_{in}}\left(\sqrt{d_{in}/r^2}\right)$.*

Similar to the results obtained from Kaiming Initialization [11], we observe that either $\zeta = \Theta_{r,d_{out},d_{in}}\left(\sqrt{d_{out}/r^2}\right)$ or $\zeta = \Theta_{r,d_{out},d_{in}}\left(\sqrt{d_{in}/r^2}\right)$ work well independently. For all models presented in this paper, either form ensures convergence. Consequently, for all subsequent experiments, we adopt $\zeta = \Theta_{r,d_{out},d_{in}}\left(\sqrt{d_{out}/r^2}\right)$.

**Remark.** *We would like to remark that the scaling factor proposed in this subsection proves to be beneficial primarily when one adopts the learning rate typically used in full-finetuning (e.g., $1e-5$), since as LoRA-GA attempts to approximate the updates of full-finetuning. However, recent research [9] suggests that LoRA with default initialization performs much better with larger learning rates. Furthermore, tensor program analysis [40, 22] indicates that higher learning rates should be paired with smaller initialization magnitudes. Therefore, we recommend decreasing or omitting the scaling factor when training using larger learning rates (e.g., $> 1e-4$).*

## 3.4 LoRA-GA Initialization

Combining the gradient approximation and stable scale components, we propose the LoRA-GA initialization method. First, we initialize $A_{\text{init}}$ and $B_{\text{init}}$ using the solution from Theorem 3.1. Then, we determine the scaling factor $\zeta$ according to Theorem 3.2 to ensure rank and scale stability. Thus, based on Theorems 3.1 and 3.2, we propose a novel initialization method, LoRA-GA.

**LoRA-GA :** We adopt $\eta = \frac{\alpha}{\sqrt{r}}$ and $\zeta = \frac{\alpha^2}{\gamma^2}\sqrt{\frac{d_{out}}{r^2}}$, where $\gamma$ is a hyperparameter. We define the index sets $I_A = \{i \mid 1 \leq i \leq r, i \in \mathbb{N}\}$ and $I_B = \{i \mid r+1 \leq i \leq 2r, i \in \mathbb{N}\}$. Denote the singular value decomposition (SVD) of $\nabla_W \mathcal{L}$ as $\nabla_W \mathcal{L} = USV^T$. The initializations are as follows:

$$A_{\text{init}} = \frac{\sqrt[4]{d_{out}}}{\gamma} V_{[1:r]}^T, \quad B_{\text{init}} = \frac{\sqrt[4]{d_{out}}}{\gamma} U_{[r+1:2r]}, \quad W_{\text{init}} = W_0 - \eta B_{\text{init}} A_{\text{init}}$$

**Algorithm 1** LoRA-GA Initialization

---

**Require:** Model $f(\cdot)$ with $L$ layers, parameters $W$, sampled batch $B = \{x, y\}$, LoRA rank $r$, LoRA
  alpha $\alpha$, loss function $\mathcal{L}$, scale factor $\gamma$
**Ensure:** Initialized parameters $W, \eta, A, B$
 1: $\hat{y} \leftarrow f(x, W)$                                                        $\triangleright$ Forward pass
 2: $\ell \leftarrow \mathcal{L}(y, \hat{y})$
 3: $\eta \leftarrow \frac{\alpha}{\sqrt{r}}$
 4: **for** $l = L, \ldots, 1$ **do**
 5:     Compute $\nabla_{W_l} \ell$                                                   $\triangleright$ Backward for one layer
 6:     $d_{out}, d_{in} \leftarrow \text{size}(W_l)$
 7:     $U, S, V \leftarrow \text{svd}(\nabla_{W_l} \ell)$
 8:     $A_l \leftarrow V_{[1:r]} \cdot \sqrt[4]{d_{out}}/\gamma$
 9:     $B_l \leftarrow U_{[r+1:2r]} \cdot \sqrt[4]{d_{out}}/\gamma$
10:     $W_l \leftarrow W_l - \eta B_l A_l$
11:     Clear $\nabla_{W_l} \ell$                                 $\triangleright$ Gradient for this layer is not needed anymore
12: **end for**
13: **return** $W, \eta, A, B$

---

To save GPU memory during LoRA-GA initialization, we utilized a technique similar to [41]. By hooking into PyTorch's backward process, we compute the gradient for one layer at a time and discard the computed gradients immediately. This ensures that our memory usage remains at $O(1)$ instead of $O(L)$, where $L$ is the number of layers. This approach allows the memory consumption during the initialization phase to be less than that during the subsequent LoRA finetuning phase. Our algorithm is shown in Algorithm 1. If the sampled batch size is large, we can also use gradient accumulation to save memory further, as shown in Algorithm 2.

## 4 Experiments

In this section, we evaluate the performance of LoRA-GA on various benchmark datasets. Initially, we assess Natural Language Understanding (NLU) capabilities using a subset of the GLUE dataset [14] with the T5-Base model [15]. Subsequently, we evaluate dialogue [16, 42], mathematical reasoning [17, 43], and coding abilities [18, 44] using the Llama 2-7B model [19]. Finally, we do the ablation study to prove the effectiveness of our method.

**Baselines**   We compare LoRA-GA with several baselines to demonstrate its effectiveness:

**1.** *Full-Finetune*: Fine-tuning the model with all parameters, which requires the most resources.
**2.** *Vanilla LoRA* [4]: Fine-tuning the model by inserting a low-rank matrix product $BA$ into linear layers. $A$ is initialized using Kaiming initialization, while $B$ is initialized to zero.
**3.** *LoRA Variants with Original Structure*: This includes several methods that retain the original LoRA structure:
    - *rsLoRA* [37] introduces a new scaling factor to stabilize the scale of LoRA.
    - *LoRA+* [36] updates the two matrices in LoRA with different learning rates.
    - *PiSSA* [38] proposes performing SVD on the weight matrix $W$ at the beginning of training and initializing $A$ and $B$ based on the components with larger singular values.
**4.** *LoRA Variants with Modified Structure*: This includes methods that modify the original LoRA structure:
    - *DoRA* [8] enhances the model's expressiveness by adding learnable magnitudes.
    - *AdaLoRA* [32] dynamically prunes insignificant weights during fine-tuning using SVD, allowing more rank allocation to important areas within a fixed parameter budget.

### 4.1   Experiments on Natural Language Understanding

**Models and Datasets**   We fine-tune the T5-Base model on several datasets from the GLUE benchmark, including MNLI, SST-2, CoLA, QNLI, and MRPC. Performance is evaluated on the development set using accuracy as the primary metric.

Table 1: Results of fine-tuning T5-base using Full-FT and various LoRA variants on a subset of GLUE.

| | MNLI | SST-2 | CoLA | QNLI | MRPC | Average |
| Size | 393k | 67k | 8.5k | 105k | 3.7k | |
| --- | --- | --- | --- | --- | --- | --- |
| Full | $86.33_{\pm 0.00}$ | $94.75_{\pm 0.21}$ | $80.70_{\pm 0.24}$ | $93.19_{\pm 0.22}$ | $84.56_{\pm 0.73}$ | 87.91 |
| LoRA | $85.30_{\pm 0.04}$ | $94.04_{\pm 0.11}$ | $69.35_{\pm 0.05}$ | $92.96_{\pm 0.09}$ | $68.38_{\pm 0.01}$ | 82.08 |
| PiSSA | $85.75_{\pm 0.07}$ | $94.07_{\pm 0.06}$ | $74.27_{\pm 0.39}$ | $93.15_{\pm 0.14}$ | $76.31_{\pm 0.51}$ | 84.71 |
| rsLoRA | $85.73_{\pm 0.10}$ | $\mathbf{94.19}_{\pm 0.23}$ | $72.32_{\pm 1.12}$ | $93.12_{\pm 0.09}$ | $52.86_{\pm 2.27}$ | 79.64 |
| LoRA+ | $\mathbf{85.81}_{\pm 0.09}$ | $93.85_{\pm 0.24}$ | $77.53_{\pm 0.20}$ | $93.14_{\pm 0.03}$ | $74.43_{\pm 1.39}$ | 84.95 |
| DoRA | $85.67_{\pm 0.09}$ | $94.04_{\pm 0.53}$ | $72.04_{\pm 0.94}$ | $93.04_{\pm 0.06}$ | $68.08_{\pm 0.51}$ | 82.57 |
| AdaLoRA | $85.45_{\pm 0.11}$ | $93.69_{\pm 0.20}$ | $69.16_{\pm 0.24}$ | $91.66_{\pm 0.05}$ | $68.14_{\pm 0.28}$ | 81.62 |
| LoRA-GA | $85.70_{\pm 0.09}$ | $94.11_{\pm 0.18}$ | $\mathbf{80.57}_{\pm 0.20}$ | $\mathbf{93.18}_{\pm 0.06}$ | $\mathbf{85.29}_{\pm 0.24}$ | $\mathbf{87.77}$ |

**Implementation Details** We utilize prompt tuning to fine-tune the T5-Base model on the GLUE benchmark. This involves converting labels into tokens (e.g., "positive" or "negative") and using the normalized probability of these tokens as the predicted label probability for classification. We provide the hyperparameters in Appendix D.1. Each experiment is conducted with 3 different random seeds, and the average performance is reported.

**Results** As shown in Table 1, LoRA-GA consistently outperforms the original LoRA and other baseline methods, achieving performance comparable to full fine-tuning. Notably, LoRA-GA excels on smaller datasets such as CoLA and MRPC, demonstrating its ability to converge faster and effectively utilize limited training data.

## 4.2 Experiment on Large Language Model

**Models and Datasets** To evaluate the scalability of LoRA-GA , we train Llama 2-7B on three tasks: *chat*, *math*, and *code*.

**1.** *Chat*: We train our model on a 52k subset of WizardLM [42], filtering out responses that begin with "As an AI" or "Sorry". We test our model on the MT-Bench dataset [16], which consists of 80 multi-turn questions designed to assess LLMs on multiple aspects. The quality of the responses is judged by GPT-4, and we report the first turn score.
**2.** *Math*: We train our model on a 100k subset of MetaMathQA [43], a dataset bootstrapped from other math instruction tuning datasets like GSM8K[17] and MATH [45], with higher complexity and diversity. We select data bootstrapped from the GSM8K training set and apply filtering. Accuracy is reported on the GSM8K evaluation set.
**3.** *Code*: We train our model on a 100k subset of Code-Feedback [44], a high-quality code instruction dataset, removing explanations after code blocks. The model is tested on HumanEval [18], which consists of 180 Python tasks, and we report the PASS@1 metric.

**Implementation Details** Our model is trained using standard supervised learning for language modelling. The loss for the input prompt is set to zero. Detailed hyperparameters can be found in Appendix D.2. Each experiment uses 3 different random seeds, and the average performance across these runs is reported.

**Result** Our results, as summarized in Table 2, indicate that LoRA-GA outperforms or is comparable to other methods, including full-finetuning. Specifically, LoRA-GA achieves superior performance on both the GSM8K and Human-eval datasets, underscoring its effectiveness in handling tasks with higher complexity and diversity. On MT-Bench, LoRA-GA also demonstrates competitive performance, although it slightly trails behind DoRA. Nevertheless, LoRA-GA achieves this with fewer parameters and approximately 70% of the training time required by DoRA. Additionally, as illustrated in Figure 2 (Left), our method exhibits a significantly faster convergence rate compared to Vanilla LoRA, with convergence rates comparable to those of full-finetuning.

**Effect of Rank**  We attribute the performance discrepancies on the GSM8K and Human-eval datasets, when compared to full-finetuning, primarily to the representational limitations imposed by the low-rank approximation. To address this, we experimented with higher ranks, specifically rank=32 and rank=128. Our findings reveal that LoRA-GA maintains stability across different rank settings and, in some cases, even surpasses full-finetuning performance. As shown in Figure 2 (Left), higher ranks with our initialization also result in loss curves that closely resemble those of full-finetuning.

Table 2: Results of fine-tuning Llama 2-7b using Full-FT and various LoRA variants, tested on MT-Bench, GSM8K, and Human-eval. LoRA-GA significantly outperforms Vanilla LoRA and approaches the performance of Full Finetune. Unless otherwise specified, the LoRA rank is set to 8.

|  | MT-Bench | GSM8K | Human-eval |
|---|---|---|---|
| Full | $5.56_{\pm0.09}$ | $54.20_{\pm0.42}$ | $19.87_{\pm0.57}$ |
| LoRA | $5.61_{\pm0.10}$ | $42.08_{\pm0.04}$ | $14.76_{\pm0.17}$ |
| PiSSA | $5.30_{\pm0.02}$ | $44.54_{\pm0.27}$ | $16.02_{\pm0.78}$ |
| rsLoRA | $5.25_{\pm0.03}$ | $45.62_{\pm0.10}$ | $16.01_{\pm0.79}$ |
| LoRA+ | $5.71_{\pm0.08}$ | $52.11_{\pm0.62}$ | $18.17_{\pm0.52}$ |
| DoRA | $\mathbf{5.97}_{\pm0.02}$ | $53.07_{\pm0.75}$ | $19.75_{\pm0.41}$ |
| AdaLoRA | $5.57_{\pm0.05}$ | $50.72_{\pm1.39}$ | $17.80_{\pm0.44}$ |
| LoRA-GA | $5.95_{\pm0.16}$ | $\mathbf{53.60}_{\pm0.30}$ | $\mathbf{19.81}_{\pm1.46}$ |
| LoRA-GA (Rank=32) | $5.79_{\pm0.09}$ | $55.12_{\pm0.30}$ | $20.18_{\pm0.19}$ |
| LoRA-GA (Rank=128) | $6.13_{\pm0.07}$ | $55.07_{\pm0.18}$ | $23.05_{\pm0.37}$ |

## 4.3 Ablation Study

We conducted ablation studies to evaluate the contributions of non-zero initialization, stable output, and gradient approximation in LoRA-GA using five distinct experimental settings. Details of each setting are provided in Table 3.

Table 3: Initialization Methods and Corresponding Settings for Ablation Study. The table compares different initialization methods for LoRA and their settings for $A$, $B$, and $\eta$. "+SO" denotes stable output, scaling parameters appropriately to ensure stability. "+GA" refers to gradient approximation, where $A$ and $B$ are initialized using orthogonal matrices derived from singular value decomposition.

| Method | $A$ Initialization | $B$ Initialization | $\eta$ |
|---|---|---|---|
| LoRA | $U\left(-\sqrt{\frac{3}{d_{in}}}, \sqrt{\frac{3}{d_{in}}}\right)$ | $0$ | $\alpha/r$ |
| Gaussian | $N(0, \frac{1}{d_{out}})$ | $N(0, \frac{1}{d_{in}})$ | $\alpha/r$ |
| +SO | $\sqrt[4]{d_{out}}/\sqrt{\gamma} \cdot N(0, \frac{1}{d_{out}})$ | $\sqrt[4]{d_{out}}/\sqrt{\gamma} \cdot N(0, \frac{1}{d_{in}})$ | $\alpha/\sqrt{r}$ |
| +GA | $V_{[1:r]}$ | $U_{[r+1:2r]}$ | $\alpha/r$ |
| LoRA-GA | $V_{[1:r]} \cdot \sqrt[4]{d_{out}}/\sqrt{\gamma}$ | $U_{[r+1:2r]} \cdot \sqrt[4]{d_{out}}/\sqrt{\gamma}$ | $\alpha/\sqrt{r}$ |

Table 4: Performance of different settings in the ablation study. Results are shown for MT-Bench, GSM8K, and Human-eval on Llama 2 7b, as well as the average performance on a subset of GLUE on T5-Base. Detailed results can be found in Table 9.

|  | MT-Bench | GSM8K | Human-eval | Average of GLUE |
|---|---|---|---|---|
| Full | $5.56_{\pm0.09}$ | $54.20_{\pm0.42}$ | $19.87_{\pm0.57}$ | 87.91 |
| LoRA | $5.61_{\pm0.10}$ | $42.08_{\pm0.04}$ | $14.76_{\pm0.17}$ | 82.08 |
| Gaussian | $5.62_{\pm0.11}$ | $38.21_{\pm0.06}$ | $14.76_{\pm0.68}$ | 81.88 |
| + SO | $5.72_{\pm0.04}$ | $42.81_{\pm1.14}$ | $15.55_{\pm0.78}$ | 82.28 |
| + GA | $5.48_{\pm0.02}$ | $46.65_{\pm1.17}$ | $16.15_{\pm0.78}$ | 82.54 |
| LoRA-GA | $5.95_{\pm0.16}$ | $53.60_{\pm0.30}$ | $19.81_{\pm1.46}$ | 87.77 |

**Ablation Result** The results are presented in Tables 4 and 9. For both small and large models, we observe that simply changing LoRA's initialization to Gaussian does not yield any performance gains and may result in a slight performance decline. However, when combined with either "+SO" (Stable Output) or "+GA" (Gradient Approximation), performance improves upon that of LoRA. LoRA-GA, which integrates both techniques, outperforms other methods. As shown in Figure 2 (Left) and Figure 4, +SO and +GA also enhance convergence speed, and when both are combined, the training loss curve is even closer to that of full-finetuning. This indicates that both output stability and gradient approximation contribute to the improvement of LoRA, each addressing different aspects of the model's performance.

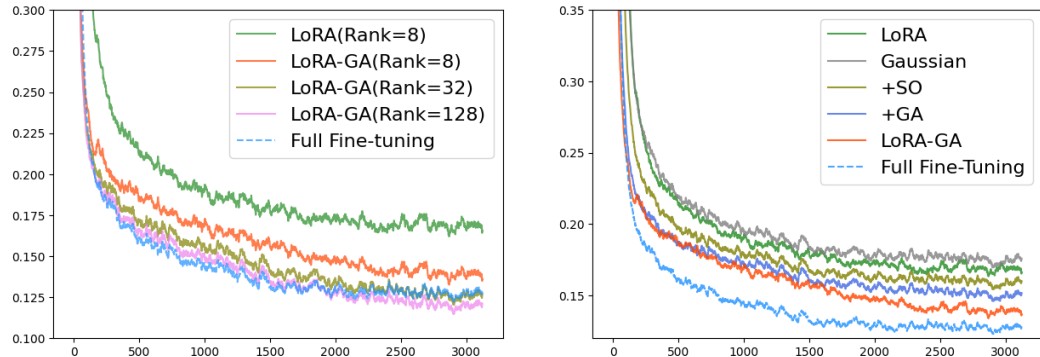

Figure 2: **(Left)** Training loss curves of LoRA-GA with different ranks on the MetaMathQA dataset. Higher ranks result in faster loss reduction, approaching the performance of full fine-tuning. **(Right)** Training loss curves from the ablation study with different settings on the MetaMATHQA dataset. Compared to Vanilla LoRA, both components of LoRA-GA , +SO (stable output) and +GA (gradient approximation), improve convergence speed. LoRA-GA achieves the fastest convergence, closely matching that of full fine-tuning.

## 4.4 Memory Costs and Running Time

We benchmark LoRA-GA on a single RTX 3090 24GB GPU, a 128-core CPU, and 256GB of RAM. As shown in Table 5, the memory consumption of our new method does not exceed that used for training with LoRA, indicating no extra memory is needed. Additionally, the time cost of this operation is relatively negligible compared to the subsequent fine-tuning process. For instance, in the Code-Feedback task, the training process took approximately 10 hours, while the initialization required only about 1 minute, which is insignificant.

Table 5: Memory and Time Costs for Initialization and Fine-Tuning. "Parameters" indicates the number of parameters in the model, "Time(LoRA-GA)" represents the time required for initialization, "Memory(LoRA-GA)" shows the memory usage during initialization, "LoRA" and "Full-FT" display the memory usage during LoRA and full fine-tuning, respectively.

|  | Parameters | Time(LoRA-GA) | Memory(LoRA-GA) | LoRA | Full-FT |
|---|---|---|---|---|---|
| T5-Base | 220M | 2.8s | 1.69G | 2.71G | 3.87G |
| Llama 2-7B | 6738M | 74.7s | 18.77G | 23.18G | 63.92G |

## 4.5 Performance with Different Index Set Schemas

Theorem 3.1 establishes multiple optimal initialization schemes through different choices of index sets $I_A$ and $I_B$. While our primary experiments employed $I_A = 1, \ldots, r$ and $I_B = r + 1, \ldots, 2r$, we conducted additional experiments to validate this choice by comparing three schemes:

- **ArB2r**: $I_A = \{1, \ldots, r\}, I_B = \{r + 1, \ldots, 2r\}$
- **A2rBr**: $I_A = \{r + 1, \ldots, 2r\}, I_B = \{1, \ldots, r\}$

- **Random**: Random assignment of first $2r$ indices into two groups

Table 6: Performance comparison of initialization schemes on GSM8k using models trained on MetaMathQA subset.

|  | ArB2r | A2rBr | Random |
|---|---|---|---|
| Performance | 52.79 | 52.38 | 52.01 |

As shown in Table 6, ArB2r slightly outperforms the alternatives. While Theorem 3.1 proves these schemas are equivalent in the first step, their behaviors diverge afterward. The gradient of matrix $B$ ($\nabla_B \mathcal{L} = (\nabla_W \mathcal{L}) A^T$) becomes larger than that of $A$ ($\nabla_A \mathcal{L} = B^T \nabla_W \mathcal{L}$), effectively increasing $B$'s learning rate. This aligns with findings from LoRA+[36], where larger learning rates for $B$ proved beneficial, potentially explaining ArB2r's superior performance.

### 4.6 Impact of Sampled Batch Size

The gradient approximation in LoRA-GA uses sampled batches, with smaller batches resembling Stochastic Gradient Descent (SGD) and larger ones approximating full Gradient Descent (GD). While theoretical work [46] suggests SGD's slower convergence may offer better generalization than GD, we conduct experiments to empirically evaluate different batch sizes.

We assess gradient approximation quality by comparing gradients from various batch sizes against a reference batch size of 2048 which serves as a proxy for the full dataset gradient using two metrics:

- **Sign Similarity**: The proportion of parameters sharing the same gradient sign.
- **Magnitude Similarity**: The proportion of parameters within the same order of magnitude (where one's absolute value is not more than 10 times the other).

Table 7: Gradient similarity metrics (vs. batch size 2048) and model performance on GSM8k using models trained on MetaMathQA subset.

| Batch Size | 8 | 16 | 32 | 64 | 128 | 256 |
|---|---|---|---|---|---|---|
| Sign Similarity | 0.743 | 0.790 | 0.838 | 0.875 | 0.903 | 0.925 |
| Magnitude Similarity | 0.878 | 0.908 | 0.933 | 0.950 | 0.962 | 0.971 |
| Performance | 52.79 | 52.99 | 52.91 | 53.56 | 52.57 | 53.22 |

As shown in Table 7, both similarity metrics consistently improve with larger batch sizes, indicating better approximation of the full gradient. The results also demonstrate that while larger batch sizes tend to yield marginally better performance, however, the differences are relatively small. Based on these findings, we recommend using a moderately large batch size (e.g., 64) when computational resources permit.

## 5 Conclusions

In this paper, we present a novel initialization scheme for low-rank adaptation (LoRA), with the goal of acelerating its convergence. By examining the initialization methods and update processes of LoRA, we develop a new initialization method, LoRA-GA , which approximates the gradients of the low-rank matrix product with those of full fine-tuning from the very first step.

Through extensive experiments, we have demonstrated that LoRA-GA achieves a convergence rate comparable to that of full fine-tuning while delivering similar or even superior performance. Since LoRA-GA solely modifies the initialization of LoRA without altering the architecture or training algorithms, it offers an efficient and effective approach that is easy to implement. Furthermore, it can also be incorporated with other LoRA variants. For example, ReLoRA [35] periodically merges the adapters into frozen weights $W$, which may allow LoRA-GA to demonstrate its advantages over more steps. We leave it as an interesting future direction.

## Acknowledgments and Disclosure of Funding

The authors are supported in part by the National Natural Science Foundation of China Grant 62161146004.

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

# A Proofs of Theorems

## A.1 Proof of Theorem 3.1

**Lemma 3.1.** *Suppose the loss function is $\mathcal{L}$ and $y = W'x = (W_0 + \eta BA)x$, where $y$ is the output of a layer and $x$ is the input, the gradients of adapters $A$ and $B$ are linear mappings of the gradient of $W'$:*

$$\nabla_A \mathcal{L} = B^T \nabla_{W'} \mathcal{L}, \quad \nabla_B \mathcal{L} = (\nabla_{W'} \mathcal{L}) A^T$$

*Remarkably, the gradient of $W'$ in LoRA and the gradient of $W$ in full fine-tuning are equal at the beginning of the training.*

*Proof.* For the gradients in LoRA,

$$\nabla_{W'} \mathcal{L} = \frac{\partial \mathcal{L}}{\partial W'} = \frac{\partial \mathcal{L}}{\partial y} \frac{\partial y}{\partial W'} = \frac{\partial \mathcal{L}}{\partial y} x^T$$

$$\nabla_A \mathcal{L} = \frac{\partial \mathcal{L}}{\partial A} = \frac{\partial W'}{\partial A} \cdot \frac{\partial \mathcal{L}}{\partial y} \frac{\partial y}{\partial W'} = B^T \cdot \frac{\partial \mathcal{L}}{\partial y} x^T = B^T \nabla_{W'} \mathcal{L}$$

$$\nabla_B \mathcal{L} = \frac{\partial \mathcal{L}}{\partial B} = \frac{\partial \mathcal{L}}{\partial y} \frac{\partial y}{\partial W'} \cdot \frac{\partial W'}{\partial B} = \frac{\partial \mathcal{L}}{\partial y} x^T A^T = (\nabla_{W'} \mathcal{L}) A^T$$

At the beginning of training, both LoRA and full fine-tuning have $y' = y$ and identical $x$, therefore,

$$\nabla_W \mathcal{L} = \nabla_{W'} \mathcal{L} = \frac{\partial \mathcal{L}}{\partial y}(y) x^T$$

$\square$

**Theorem 3.1.** *Consider the following optimization problem:*

$$\min_{A_{\text{init}}, B_{\text{init}}} \left\| \eta^2 \nabla_W \mathcal{L} \cdot A_{\text{init}}^T A_{\text{init}} + \eta^2 B_{\text{init}} B_{\text{init}}^T \cdot \nabla_W \mathcal{L} - \zeta \nabla_W \mathcal{L} \right\|_F$$

*If the Singular Value Decomposition (SVD) of $\nabla_W \mathcal{L}$ is $\nabla_W \mathcal{L} = USV^T$, the solution to this optimization problem is:*

$$B_{\text{init}} = \frac{\sqrt{\zeta}}{\eta} U_{I_A}, \quad A_{\text{init}} = \frac{\sqrt{\zeta}}{\eta} V_{I_B}^T \quad s.t. \ |I_A| = |I_B| = r, \ I_A \cup I_B = \{i \mid 1 \le i \le 2r, i \in \mathbb{N}\}$$

*where $I_A, I_B$ are index sets.*

*Proof.* Since that $rank(A_{\text{init}}) = rank(B_{\text{init}}) = r$ and $2r < \min(m, n)$, we can assert that the matrix $W' = \eta^2 \nabla_W \mathcal{L} A_{init}^T A_{init} + \eta^2 B_{init} B_{init}^T \nabla_W \mathcal{L}$ has $rank(W') \le 2r$.

Under this given solution,

$$W' = \eta^2 \nabla_W \mathcal{L} A_{\text{init}}^T A_{\text{init}} + \eta^2 B_{\text{init}} B_{\text{init}}^T \nabla_W \mathcal{L} = \zeta USV^T (V_{I_A} V_{I_A}^T) + \zeta (U_{I_B} U_{I_B}^T) USV^T$$

$$= \zeta \sum_{i \in I_A} \sigma_i u_i v_i^T + \zeta \sum_{j \in I_B} \sigma_j u_j u_j^T = \zeta \sum_{i=1}^{2r} \sigma_i u_i v_i^T$$

By the classic Eckart-Young Theorem (see e.g., [47, 48]), the optimal low-rank approximation with respect to Frobenius norm is:

$$W'^* = \arg \min_{rank(W'^*)=2r} \left\| W'^* - \zeta \nabla_W \mathcal{L} \right\|_F = \zeta \sum_{i=1}^{2r} \sigma_i u_i v_i^T$$

This is identical to what we have got. Therefore, this is the optimal solution. $\square$

## A.2 Proof of Theorem 3.2

**Lemma A.1.** *In $\mathbb{R}^n$, if we randomly pick a vector $x$ that $\sum_{i=1}^n x_i^2 = 1$, we have:*

1. $\mathbb{E}(x_i) = 0$, $\mathbb{E}(x_i^2) = \frac{1}{n}$ and $\mathbb{E}(x_i^4) = \Theta_{r, d_{out}, d_{in}}(\frac{1}{n^2})$;

2. $\mathbb{E}(x_i x_j) = 0$;

3. $\mathbb{E}(x_i^2 x_j^2) = \Theta_{r, d_{out}, d_{in}}(\frac{1}{n^2})$;

4. $\mathbb{E}\left(x_i^2 x_j x_k\right) = 0$;

*Proof.* It is equivalent to sampling a random point uniformly from a unit sphere in $\mathbb{R}^n$.

For property 1, $\mathbb{E}\left(x_i\right) = 0$ holds obvious by symmetry. Since $\sum_{i=1}^n x_i^2 = 1$ and uniformly distributed, each entry has identical expectation, $\mathbb{E}\left(\sum_{i=1}^n x_i^2\right) = n\mathbb{E}\left(x_i^2\right) = 1$, $\mathbb{E}\left(x_i^2\right) = \frac{1}{n}$. $\mathbb{E}\left(x_i^4\right) = \mathbb{E}\left(x_i^2 \cdot x_i^2\right) = \Theta_{r,d_{out},d_{in}}\left(\frac{1}{n}\right)\Theta_{r,d_{out},d_{in}}\left(\frac{1}{n}\right) = \Theta_{r,d_{out},d_{in}}\left(\frac{1}{n^2}\right)$.

For property 2, it can also be proved by symmetry: we can always find vector that contains $(x_i, -x_j)$ also lies on the sphere. Therefore, $\mathbb{E}\left(x_i x_j\right) = 0$.

For property 3, $\mathbb{E}\left(x_i^2 x_j^2\right) = \mathbb{E}\left(x_i^2 \cdot x_j^2\right) = \Theta_{r,d_{out},d_{in}}\left(\frac{1}{n}\right)\Theta_{r,d_{out},d_{in}}\left(\frac{1}{n}\right) = \Theta_{r,d_{out},d_{in}}\left(\frac{1}{n^2}\right)$.

For property 4, again it can be proved by symmetry: we can always find vector that contains $(x_i, x_j, -x_k)$ also lies on the sphere. Therefore, $\mathbb{E}\left(x_i^2 x_j x_k\right) = 0$. $\qquad\square$

**Lemma A.2.** *For a randomly selected orthogonal matrix $A \in \mathbb{R}^{n \times n}$, and we randomly pick two different column vectors $x$ and $y$ from it. For these two vectors, we have the following:*

1. $\mathbb{E}\left(x_i y_i\right) = 0$;

2. $\mathbb{E}\left(x_i y_j\right) = 0$;

*Proof.* It is equivalent to first selecting a random vector $x$ from a unit sphere in $\mathbb{R}^n$ uniformly, and then selecting the other one $y$ that is orthogonal to $x$.

For property 1, $\sum_{i=1}^n x_i y_i = 0 \Rightarrow \mathbb{E}\left(\sum_{i=1}^n x_i y_i\right) = \sum_{i=1}^n \mathbb{E}\left(() x_i y_i\right) = 0 \Rightarrow \mathbb{E}\left(x_i y_i\right) = 0$.

For property 2, consider that $\mathbb{E}\left(\sum_{i=1}^n x_i\right) = \mathbb{E}\left(\sum_{i=1}^n y_i\right) = 0$, and given $x$, we can always find $-y$ is also an orthogonal vector. Therefore, $\mathbb{E}\left(\sum_{i=1}^n x_i \sum_{i=1}^n y_i\right) = 0 \Rightarrow E(x_i y_i) = 0$. $\qquad\square$

**Theorem 3.2.** *Given the initialization proposed in Theorem 3.1, assume that the orthogonal vectors in $A_{\text{init}}$ and $B_{\text{init}}$ are randomly selected from $\mathbb{R}^{d_{in}}$ and $\mathbb{R}^{d_{out}}$, and set $\eta = \Theta_{r,d_{out},d_{in}}\left(\frac{1}{\sqrt{r}}\right)$ as suggested by rsLoRA [37].*

*Under these conditions, the adapters are forward scale-stable if $\zeta = \Theta_{r,d_{out},d_{in}}\left(\sqrt{\frac{d_{out}}{r^2}}\right)$ and backward scale-stable if $\zeta = \Theta_{r,d_{out},d_{in}}\left(\sqrt{\frac{d_{in}}{r^2}}\right)$.*

*Proof.* In LoRA, $h = (W' + \eta BA)x$, since that $W'$ is not considered here, therefore, denote $y = \eta BAx$. When backward propagation, it's like $\frac{\partial \mathcal{L}}{\partial x} = \eta A^T B^T \frac{\partial \mathcal{L}}{\partial h}$. Represente $\frac{\partial \mathcal{L}}{\partial h}$ as $v$ and $\frac{\partial \mathcal{L}}{\partial x}$ as $g$. Therefore,

$$y_i = \eta \sum_{j=1}^r \sum_{k=1}^{d_{in}} B_{ij} A_{jk} x_k, \ 1 \le i \le d_{out} \quad \text{(Forward)}$$

$$g_i = \eta \sum_{j=1}^r \sum_{k=1}^{d_{out}} A_{ji} B_{kj} v_k, \ 1 \le i \le d_{in} \quad \text{(Backward)}$$

(4)

Since that the output of each layer in model always passes a softmax function, so that the vector $\frac{\partial \mathcal{L}}{\partial h} = v$ is $\Theta_{r,d_{out},d_{in}}\left(1\right)$. Further, since that input $x_i$'s are *i.i.d.*, without loss of generality, assume that $E(x_i) = 0$ and $E(x_i^2) = 1$.

For the adapter, as Equation 4 shows, and by the expectations we have proved in Lemma A.1 and A.2, we can calculate the scale of forward and backward process.

The scale of forward process is:

$$
\begin{aligned}
\mathbb{E}\left(y_i^2\right) &= \eta^2 \sum_{j_1=1}^{r} \sum_{j_2=1}^{r} \sum_{k_1=1}^{d_{in}} \sum_{k_2=1}^{d_{in}} \mathbb{E}\left(B_{ij_1} A_{j_1 k_1} B_{ij_2} A_{j_2 k_2} x_{k_1} x_{k_2}\right) \\
&= \eta^2 \sum_{j_1=1}^{r} \sum_{j_2=1}^{r} \sum_{k_1=1}^{d_{in}} \sum_{k_2=1}^{d_{in}} \mathbb{E}\left(B_{ij_1} B_{ij_2}\right) \mathbb{E}\left(A_{j_1 k_1} A_{j_2 k_2}\right) \mathbb{E}\left(x_{k_1} x_{k_2}\right) \\
&= \eta^2 \sum_{j_1=1}^{r} \sum_{j_2=1}^{r} \sum_{k=1}^{d_{in}} \mathbb{E}\left(B_{ij_1} B_{ij_2}\right) \mathbb{E}\left(A_{j_1 k} A_{j_2 k}\right) = \eta^2 \sum_{j=1}^{r} \sum_{k=1}^{d_{in}} \mathbb{E}\left(B_{ij}^2\right) \mathbb{E}\left(A_{jk}^2\right) \\
&= \eta^2 \sum_{j=1}^{r} \sum_{k=1}^{d_{in}} \frac{\zeta^2}{\eta^4} \frac{1}{d_{out}} \frac{1}{d_{in}} = \frac{1}{\alpha^2} \cdot \zeta^2 \cdot \frac{r^2}{d_{out}}
\end{aligned}
\tag{5}
$$

The scale of the backward process is:

$$
\begin{aligned}
\mathbb{E}\left(g_i^2\right) &= \eta^2 \sum_{j_1=1}^{r} \sum_{j_2=1}^{r} \sum_{k_1=1}^{d_{out}} \sum_{k_2=1}^{d_{out}} \mathbb{E}\left(A_{j_1 i} B_{k_1 j_1} A_{j_2 i} B_{k_2 j_2} v_{k_1} v_{k_2}\right) \\
&= \eta^2 \sum_{j_1=1}^{r} \sum_{j_2=1}^{r} \sum_{k_1=1}^{d_{out}} \sum_{k_2=1}^{m} v_{k_1} v_{k_2} \mathbb{E}\left(A_{j_1 i} A_{j_2 i}\right) \mathbb{E}\left(B_{k_1 j_1} B_{k_2 j_2}\right) \\
&= \eta^2 \sum_{j=1}^{r} \sum_{k=1}^{d_{out}} v_k^2 \mathbb{E}\left(A_{ji}^2\right) \mathbb{E}\left(B_{kj}^2\right) = \eta^2 \sum_{j=1}^{r} \sum_{k=1}^{d_{out}} v_k^2 \frac{\zeta^2}{\eta^4} \frac{1}{d_{in}} \frac{1}{d_{out}} = \frac{1}{\alpha^2} \cdot \zeta^2 r^2 \Theta_{r, d_{out}, d_{in}}\left(\frac{1}{d_{in}}\right)
\end{aligned}
\tag{6}
$$

From the results derived by Equation 5 and 6, one can see that we cannot find a proper $\zeta$ to make both scales $\Theta_{r, d_{out}, d_{in}}(1)$ unless $\frac{d_{out}}{d_{in}} = \Theta_{r, d_{out}, d_{in}}(1)$. We can also see that the forward scale is stable if adopting $\zeta = \Theta_{r, d_{out}, d_{in}}\left(\frac{d_{out}}{r^2}\right)$ and the backward is stable if $\zeta = \Theta_{r, d_{out}, d_{in}}\left(\frac{d_{in}}{r^2}\right)$. □

# B Additional Experimental Results

## B.1 Convergence Speed

As Figure 3 and 4 shown, the convergence of LoRA-GA is significantly faster than vanilla LoRA and other ablation models, almost close to that of full fine-tuning, which support our claim about the speed of convergence.

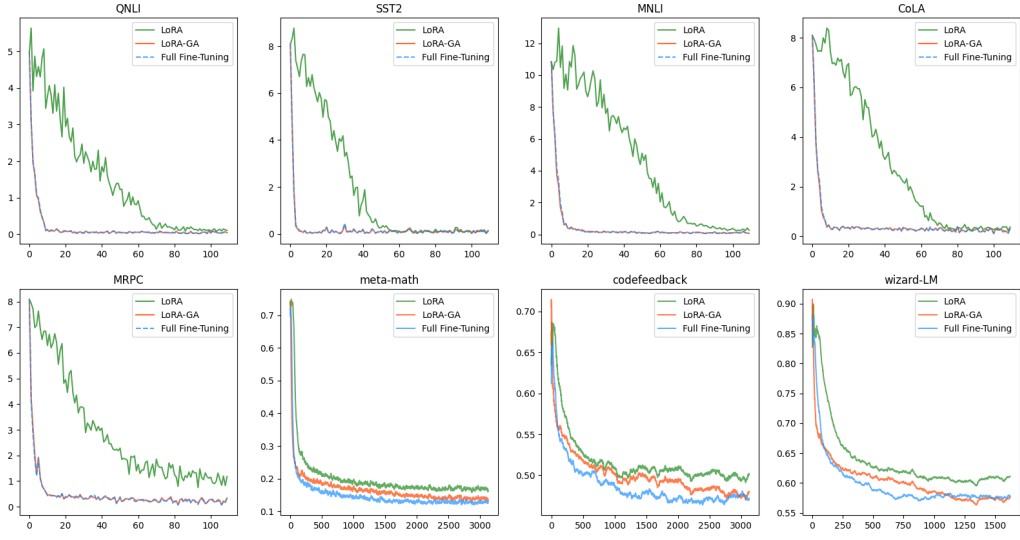

Figure 3: Training Loss curves of Full Fine-tuning, LoRA and LoRA-GA on different datasets.

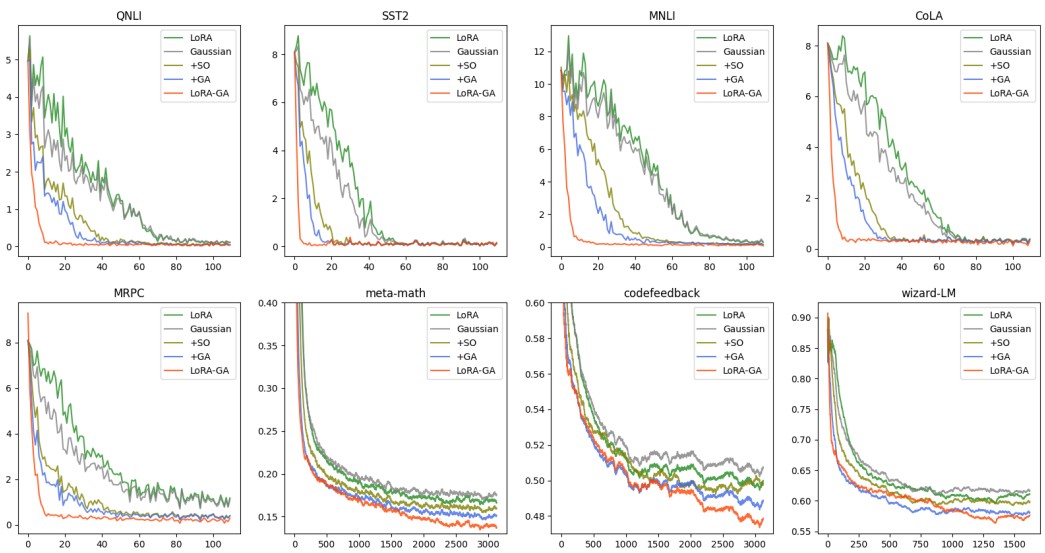

Figure 4: Training Loss curves of different LoRA-GA ablations on different datasets.

## B.2 Evaluating the Rank of the Gradient Matrix

Theorem 3.1 suggests that the closer the rank of the gradient matrix is to $2r$, the better the gradient approximated, thereby enhancing the theoretical effectiveness of our initialization. Figure 5 illustrates the low-rank nature of gradient matrices. The left panel depicts a grid-like pattern in the gradients of a weight matrix, indicating a low-rank structure. The middle panel shows a steeply declining curve of singular values, reflecting the highly low-rank nature of the gradient matrix. The right panel presents the cumulative curve of squared singular values, demonstrating that a few ranks account for nearly all the singular values of the gradient matrix. Specifically, the coverage in the right panel is defined as

$$\text{Coverage} = \frac{\sum_{i=0}^{2r} \sigma_i^2}{\sum_{i=0}^{n} \sigma_i^2},$$

where $r$ is the LoRA rank used in LoRA-GA , indicating how much of the low-rank matrix can be approximated by this rank.

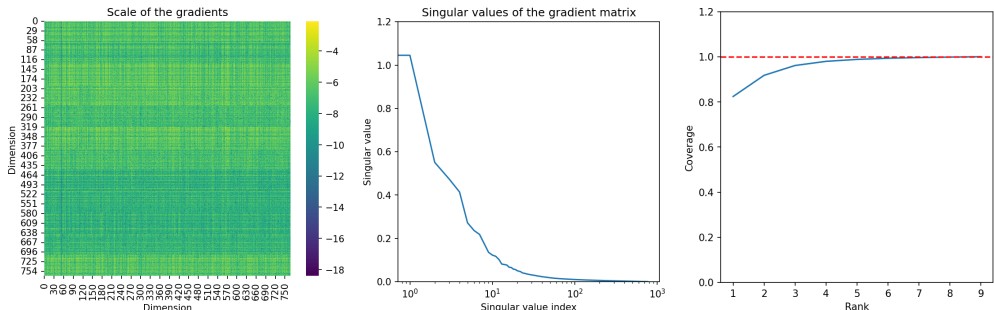

Figure 5: (**Left**) A gradient matrix of T5-Base during fine-tuning on CoLA. (**Middle**) The decreasing curve of singular values of the gradient matrix. (**Right**) The cumulative curve showing the coverage of squared singular values.

We further validate this observation on larger models by analyzing LLaMA 2-7B during MetaMathQA training. Table 8 presents the coverage across different layers with varying LoRA ranks. Even with a relatively small rank of 8, we achieve a mean coverage of 92.9% across all layers, with the minimum coverage being 85.1%. Increasing the rank to 128 yields an impressive mean coverage of 99.3%, with the minimum coverage reaching 97.5%. These results demonstrate that even for large models with weight matrices of dimension 4096, a modest LoRA rank is sufficient to capture the majority of the gradient information.

Table 8: Coverage of gradient matrix across different layers in LLaMA 2-7B

| LoRA Rank | 8 | 32 | 128 |
|---|---|---|---|
| Mean Coverage | 0.929 | 0.974 | 0.993 |
| Min Coverage | 0.851 | 0.933 | 0.975 |

## B.3 Detailed Ablation Study Result of GLUE

Table 9 shows the full results of ablation study on the subset of GLUE, where the average scores are briefly reported in Table 4. As Table 9 demonstrated, LoRA-GA outperforms all other ablation models, while both "+SO" and "+GA" methods gain some improvement from vanilla LoRA and simple non-zero initialization "Gaussian". This illustrates that both components in LoRA-GA have positive contribution to the improvement of performance.

Table 9: Performance comparison of different ablations on subset of GLUE dataset. The settings are elaborated in Table 3.

| Trainset | MNLI 393k | SST-2 67k | CoLA 8.5k | QNLI 105k | MRPC 3.7k | Average |
|---|---|---|---|---|---|---|
| Full | $86.33_{\pm 0.00}$ | $94.75_{\pm 0.21}$ | $80.70_{\pm 0.24}$ | $93.19_{\pm 0.22}$ | $84.56_{\pm 0.73}$ | 87.91 |
| LoRA | $85.30_{\pm 0.04}$ | $94.04_{\pm 0.11}$ | $69.35_{\pm 0.05}$ | $92.96_{\pm 0.09}$ | $68.38_{\pm 0.01}$ | 82.08 |
| Gaussian | $85.26_{\pm 0.07}$ | $93.85_{\pm 0.18}$ | $69.00_{\pm 0.22}$ | $92.89_{\pm 0.08}$ | $\underline{68.38}_{\pm 0.00}$ | 81.88 |
| + SO | $\underline{85.47}_{\pm 0.19}$ | $\mathbf{94.23}_{\pm 0.13}$ | $70.63_{\pm 0.78}$ | $\underline{93.12}_{\pm 0.07}$ | $67.97_{\pm 0.75}$ | 82.28 |
| + GA | $85.33_{\pm 0.07}$ | $93.88_{\pm 0.18}$ | $\underline{74.37}_{\pm 1.12}$ | $93.03_{\pm 0.06}$ | $66.09_{\pm 11.32}$ | 82.54 |
| LoRA-GA | $\mathbf{85.70}_{\pm 0.09}$ | $\underline{94.11}_{\pm 0.18}$ | $\mathbf{80.57}_{\pm 0.20}$ | $\mathbf{93.18}_{\pm 0.06}$ | $\mathbf{85.29}_{\pm 0.24}$ | **87.77** |

## B.4 Experimental result with different learning rate

Furthermore, we also conduct experiments under learning rates 1e-5 and 5e-5. As Table 10 and 11 shown, LoRA-GA maintains strong performance across different learning rates, which illustrating its robustness to the variation of learning rate.

Table 10: Performance comparison of different methods on MT-Bench, GSM8K, and Human-eval with learning rate 1e-5

| | MT-Bench | GSM8K | Human-eval |
|---|---|---|---|
| Full | $5.63_{\pm 0.04}$ | $43.95_{\pm 1.95}$ | $15.97_{\pm 0.42}$ |
| LoRA | $5.53_{\pm 0.07}$ | $35.73_{\pm 0.09}$ | $14.35_{\pm 0.40}$ |
| PiSSA | $5.61_{\pm 0.09}$ | $38.51_{\pm 0.70}$ | $15.37_{\pm 0.78}$ |
| rsLoRA | $5.60_{\pm 0.10}$ | $40.56_{\pm 0.47}$ | $15.69_{\pm 0.87}$ |
| LoRA+ | $5.48_{\pm 0.14}$ | $47.06_{\pm 0.11}$ | $16.90_{\pm 0.89}$ |
| LoRA-GA | $\mathbf{5.82}_{\pm 0.04}$ | $\mathbf{51.33}_{\pm 0.39}$ | $\mathbf{17.64}_{\pm 0.13}$ |

Table 11: Performance comparison of different methods on MT-Bench, GSM8K, and Human-eval with learning rate 5e-5

| | MT-Bench | GSM8K | Human-eval |
|---|---|---|---|
| Full | $5.33_{\pm 0.21}$ | $56.33_{\pm 0.78}$ | $25.67_{\pm 0.42}$ |
| LoRA | $5.52_{\pm 0.08}$ | $46.89_{\pm 0.05}$ | $15.67_{\pm 0.60}$ |
| PiSSA | $5.35_{\pm 0.01}$ | $49.70_{\pm 0.80}$ | $17.62_{\pm 0.60}$ |
| rsLoRA | $5.54_{\pm 0.00}$ | $50.04_{\pm 0.54}$ | $17.38_{\pm 0.26}$ |
| LoRA+ | $\mathbf{5.89}_{\pm 0.11}$ | $\mathbf{55.23}_{\pm 0.16}$ | $19.21_{\pm 0.37}$ |
| LoRA-GA | $5.76_{\pm 0.22}$ | $52.79_{\pm 1.02}$ | $\mathbf{20.45}_{\pm 0.92}$ |

## B.5 Experiments on the Full MetaMathQA Dataset

Following [9], we conducted additional experiments by training on the complete MetaMathQA dataset for multiple epochs, whereas our main results in the previous section were based on fine-tuning for one epoch on the 100k subset of MetaMathQA. Due to computational constraints, we limited these extended experiments to three methods: LoRA[4], LoRA+[36], and LoRA-GA .

Table 12 presents the performance across four epochs, averaged over two random seeds.

Table 12: Performance comparison of different methods on full MetaMathQA dataset training for multiple epochs.

|  | Epoch 1 | Epoch 2 | Epoch 3 | Epoch 4 |
|---|---|---|---|---|
| LoRA (Rank=8) | 55.19 | 58.37 | 59.28 | 58.90 |
| LoRA+ (Rank=8) | 56.37 | **59.21** | 59.93 | 59.97 |
| LoRA-GA (Rank=8) | **56.48** | 58.64 | **60.16** | **60.88** |

The results show that LoRA-GA consistently achieves better performance than vanilla LoRA and outperforms LoRA+ in most cases across multiple epochs of training.

## C LoRA-GA Initialization With Gradient Accumulation

---
**Algorithm 2** LoRA-GA Initialization With Gradient Accumulation

---
**Require:** Model $f(\cdot)$ with $L$ layers, parameters $W$, sampled batch $B = \{x, y\}$, LoRA rank $r$ with $n$ samples, LoRA alpha $\alpha$, loss function $\mathcal{L}$, scale factor $\gamma$, micro-batch size $b$
**Ensure:** Initialized parameters $W, \eta, A, B$
1: $\hat{y} \leftarrow f(x, W)$         ▷ Forward pass
2: $\ell \leftarrow \mathcal{L}(y, \hat{y})$
3: $\eta \leftarrow \frac{\alpha}{\sqrt{r}}$
4: **for** $l = 1, \ldots, L$ **do**
5:      $\nabla_{W_l}^{\text{avg}} \ell \leftarrow 0$         ▷ Initialize average gradient for each layer on CPU
6: **end for**
7: **for** each micro-batch $B_i$ in $B$ **do**
8:      $\hat{y}_i \leftarrow f(x_i, W)$         ▷ Forward pass for micro-batch
9:      $\ell_i \leftarrow \mathcal{L}(y_i, \hat{y}_i)$
10:      **for** $l = L, \ldots, 1$ **do**
11:          Compute $\nabla_{W_l} \ell_i$         ▷ Backward pass for one layer
12:          $\nabla_{W_l}^{\text{avg}} \ell \leftarrow \nabla_{W_l}^{\text{avg}} \ell + \nabla_{W_l} \ell_i \cdot \frac{b}{n}$         ▷ Move to CPU
13:          Clear $\nabla_{W_l} \ell_i$         ▷ Gradient for this layer is not needed anymore
14:      **end for**
15: **end for**
16: **for** $l = L, \ldots, 1$ **do**
17:      $d_{out}, d_{in} \leftarrow \text{size}(W_l)$
18:      $U, S, V \leftarrow \text{svd}(\nabla_{W_l}^{\text{avg}} \ell)$
19:      $A_l \leftarrow V_{[1:r]} \cdot \sqrt[4]{d_{out}}/\gamma$
20:      $B_l \leftarrow U_{[r+1:2r]} \cdot \sqrt[4]{d_{out}}/\gamma$
21:      $W_l \leftarrow W_l - \eta B_l A_l$
22: **end for**
23: **return** $W, \eta, A, B$

---

## D Hyperparameter

### D.1 Experiments on Natural Language Understanding

We use the following hyperparameters with T5-Base.

- Training Algorithm: AdamW [49] with $\beta_1 = 0.9$, $\beta_2 = 0.999$, $\epsilon = 1e - 8$ and weight decay of 0. For full finetuning, LoRA, and its variants, a learning rate of $1e - 4$ , a warmup ratio of 0.03, and cosine decay are employed. For DoRA [8], a learning rate of $2e - 4$ is used, while for Adalora, a learning rate of $5e - 4$ is applied, both with the same warmup ratio and cosine decay adhering to their respective papers.

- LoRA Hyperparameters: LoRA rank $r = 8$, $\alpha = 16$. LoRA target is all linear modules except embedding layer, layer norm and language model head.

- LoRA-GA Hyperparameter: $\gamma = 16$, sampled batch size $sbs = 8$

- Other Hyperparameters: Sequence Length $T = 128$, train batch size $bs = 32$, number of train epochs $E = 1$. Precision FP32

## D.2 Experiment on Large Language Model

We use the following hyperparameters with Llama 2-7B.

- Training Algorithm: AdamW [49] with with $\beta_1 = 0.9$, $\beta_2 = 0.999$, $\epsilon = 1e - 8$ and weight decay of 0. For full finetuning, LoRA, and its variants, a learning rate of $2e - 5$ [38], a warmup ratio of 0.03, and cosine decay are employed. For DoRA [8], a learning rate of $2e - 4$ is used, while for Adalora, a learning rate of $5e - 4$ is applied, both with the same warmup ratio and cosine decay adhering to their respective papers.

- Precision: The backbone model uses bf16 precision, while during training, LoRA's $B$ and $A$ matrices use fp32 precision, following the implementation of PEFT [39].

- LoRA-GA Hyperparameter: $\gamma = 64$, micro sampled batch size $sbs = 1$ with gradient accumulation of 32.

- LoRA Hyperparameters: LoRA rank $r = 8$ and $\alpha = 16$ for all experiments.

- Generation Hyperparameters: All generation is performed with $top\_p = 0.95$ and temperature $T = 0.8$.

- Other Hyperparameters: Number of train epochs $E = 1$, train micro batch size $mbs = 1$ with gradient accumulation of 32. Sequence Length $T = 1024$

# E    Comparison between LoRA-GA and PiSSA

Both LoRA-GA and PiSSA [38] concentrate on the initialization of LoRA, and utilizing SVD on pre-trained models. While they may appear similar superficially, significant differences exist between them.

Firstly, the motivations behind LoRA-GA and PiSSA are fundamentally different. As discussed in Section 3.2, LoRA-GA is motivated by the approximation of the LoRA update and full fine-tuning. We employ SVD on gradients solely because the optimal solution to the gradient approximation problem is precisely obtained (as stated in Theorem 3.1). Conversely, PiSSA adopts SVD under the assumption that pre-trained weights possess a low intrinsic rank, and thus, the SVD of weights can provide an accurate representation of original weights. In essence, LoRA-GA emphasizes on gradients and decomposes them, whereas PiSSA concentrates on weights and decomposes them.

Secondly, LoRA-GA and PiSSA employ different scales of initialization. In Section 3.3, LoRA-GA derives an appropriate scaling factor by considering the forward and backward stability of our initialization scheme. On the other hand, PiSSA uses the largest $r$ singular values as the magnitude of orthogonal matrices directly.

# F    Limitations

In this paper, we have demonstrated that LoRA-GA can achieve performance comparable to full fine-tuning on the T5-Base (220M) and Llama 2-7B models, while significantly reducing the number of parameters and associated costs. However, due to computational resource constraints, we have not validated LoRA-GA on larger pre-trained models (e.g., Llama 2-70B).

In LoRA-GA , we proposed that a scaling factor is necessary. But in some experiments with large learning rates, we observed potential numerical instability due to the effect of scaling factor and learning rate. This limitation suggests a need for careful tuning of the scaling factor and learning rate to maintain stability.

Another limitation pertains to our evaluation scope. While we provide evaluations on MTBench, GSM8K, and Human-eval, we did not assess our method on other datasets. Consequently, we cannot fully guarantee that our findings are universally consistent across all benchmarks.

Additionally, we did not implement our method on other LoRA variants that are orthogonal to our improvements (e.g., ReLoRA [35]). Therefore, we cannot ascertain whether LoRA-GA would perform equally well with other LoRA architectures/improvements.

Finally, compared to the original LoRA, LoRA-GA requires double the checkpoint storage, as it necessitates storing both the initial adapter checkpoints ($A_{init}$ and $B_{init}$) and the final adapter checkpoints ($A$ and $B$).

# G    Compute Resources

In this paper, we utilized two types of GPUs: the RTX 3090 24GB GPU, supported by a 128-core CPU and 256GB of RAM (hereinafter referred to as "the RTX 3090"), and the A100 80GB GPU (hereinafter referred to as "the A100").

For the experiments on T5-Base using the GLUE dataset, reported in Section 4.1, all computations were performed on a single RTX 3090. For the Llama 2-7B experiments, reported in Section 4.2, full fine-tuning and DoRA scenarios were conducted on a single A100, while all other LoRA variants and LoRA-GA were executed on a single RTX 3090. Additionally, all ablation studies presented in Section 4.3 were carried out on a single RTX 3090.

# H    Broader Impacts

In this paper, we identify some limitations of vanilla LoRA and propose a more efficient and effective method for LoRA initialization, LoRA-GA. LoRA-GA converges faster than vanilla LoRA and consistently achieves better evaluation results.

We believe that this work will have a positive social impact. The primary reasons are as follows: The high cost of training and fine-tuning large models is a significant challenge today. LoRA-GA offers a way to fine-tune with fewer parameters and lower computational costs while still achieving comparable performance. This will reduce the cost of fine-tuning models and, in turn, decrease energy consumption, such as electricity, contributing to the goal of a low-carbon environment. Furthermore, as the size of large language models (LLM) continues to grow, it becomes increasingly difficult for individuals or small organizations to develop their own LLMs. However, with the help of LoRA-GA and open-source large models, the hardware barrier to entry in this area is greatly reduced. This will promote democratization in the field of large models, preventing monopolies and dictatorships by a few companies.

On the other hand, our method could potentially make it easier to train language models that generate fake news or misleading information. This underscores the necessity for designing effective detectors to identify content generated by large language models (LLMs). Ensuring the responsible use of this technology is crucial to mitigating the risks associated with the misuse of advanced language models.

