# OpenReview forum: "LoRA-GA: Low-Rank Adaptation with Gradient Approximation"
_NeurIPS.cc/2024/Conference — NeurIPS 2024 poster_

### Official Review · Reviewer_dSFq · 2024-07-11

**Soundness:** 3
**Presentation:** 3
**Contribution:** 2
**Rating:** 6
**Confidence:** 3

**Summary:**

This paper proposes LoRA-GA, which uses an adapter to approximate the gradient update of weights. This method achieves a 2-4 times improvement in convergence speed compared to vanilla LoRA and offers better accuracy than other LoRA-based methods.

**Strengths:**

1. This paper provides a novel perspective on the initialization of LoRA from the gradient  updation aspect.
2. The paper is theoretically comprehensive, well-written, and easy to follow.

**Weaknesses:**

1. The difference between LoRA-GA and LoRA reparameterization is not shown in Figure 1.
2. A discussion on the influence of sampled batch size seems necessary.

**Questions:**

1、	The accuracy provided in Table 2 is quite different from the numbers in PiSSA [1]. I wonder where the discrepancy comes from.
2、	From Table 4, the performance improvement brought by SO to Gaussian initialization is about 0.4%, while the improvement brought by SO to GA is 5.0%. Since the SO method appears to be general in Section 3.3, why does SO perform so well with GA?
3、	How does the sampled batch size influence the initialization?
4、	What would happen if the value of r in A and B were swapped? Or if the index sets I_{A} and I_{B} were randomly chosen from [1,2r]?

[1] Pissa: Principal singular values and singular vectors adaptation of large language models[J]. arXiv.

---

> ### Author Rebuttal · Authors · 2024-08-07
>
> Thank you for your valuable questions and suggestion!
> ### Weakness
>
> **Weakness**: The difference between LoRA-GA and LoRA reparameterization is not shown in Figure 1.
>
> **Answer**: The parameterization of LoRA and LoRA-GA is identical, with both methods utilizing low-rank matrices $A$ and $B$, as depicted in Figure 1. The key (and only) distinction lies in their initialization approaches, as described in the "Initialization" section of Figure 1. We will clarify this explicitly in the next version.
>
> ### Question 1
>
> **Question 1**: The accuracy provided in Table 2 is quite different from the numbers in PiSSA [1]. I wonder where the discrepancy comes from.
>
> **Answer 1**: The discrepancy between PiSSA and our results is attributed to different rank settings. PiSSA reports accuracies on various tasks in Table 1 using a rank of 128 (according to **Figure 6(c) and Figure 14(b)** in their paper, only with rank=128 can such high performance be achieved), while our accuracy (**in Table 2**) is based on a rank of 8 for all LoRA variants. When comparing accuracies at a rank of 8, the accuracy for PiSSA that we implemented on GSM8K is 44.54 (**Table 2**), which surpasses the reported accuracy of less than 40 (**Figure 6(c) and Figure 14(b)**) in PiSSA’s paper. Remarkably, when comparing the performance at rank 128 from our paper and PiSSA, LoRA-GA still outperforms PiSSA on GSM8K (**55.07** vs. 53.07), and Human-eval (**23.05** vs. 21.95). Furthermore, even LoRA-GA at rank 8 surpasses PiSSA at rank 128 on GSM8K (**53.60** vs. 53.07).
>
> ### Question 2
>
> **Question 2**: From Table 4, the performance improvement brought by SO to Gaussian initialization is about 0.4%, while the improvement brought by SO to GA is 5.0%. Since the SO method appears to be general in Section 3.3, why does SO perform so well with GA?
>
> **Answer 2**: Thanks for noticing this interesting discrepancy. Although we admit that we have not full understood the exact reason for this discrepancy, we suspect the following reason:  The SO method adjusts both the output and update magnitude. The GA method approximates the gradient of full fine-tuning, thereby identifying a good descent direction. Intuitively, with only SO, the model might take a large step in the wrong direction initially (due to random initialization), leading to a suboptimal region and affecting later optimization. However, with SO and GA, the initial steps are more accurate, and a larger step size becomes beneficial. Hence, SO performs much better in combination of GA. Investigating the deeper reason of this fact remains open to further exploration.
>
> ### Question 3
>
> **Question 3**: How does the sampled batch size influence the initialization?
>
> **Answer 3**: LoRA-GA approximates the sampled batch gradient. Smaller batches resemble SGD, while larger batches are similar to full GD. According to Amir [1], SGD converges more slowly than GD but has better generalization. Therefore, it's difficult to definitively say which approach is theoretically superior. Hence, we conduct experiments to get empirical results for different sampled batch size:
>
> 1. The similarity between sampled batch and full gradient:
>
>    To compare the gradients, we used sampled batch sizes of 8, 16, 32, 64, 128, and 256, and compared them with a large batch size of 2048, simulating the full dataset gradient. We used two metrics:
>
>    - Sign similarity: The proportion of parameters with the same gradient sign, important because Adam's first step is similar to SignGD.
>    - Magnitude similarity: The proportion of parameters within the same magnitude level (one's absolute value not larger than 10 times of the other).
>
>    |                      | 8 | 16 | 32 | 64 | 128 | 256 |
>    | :---: | :---: | :---: | :---: | :---: | :---: | :---: |
>    |   Sign similarity    |  0.743  |  0.790  |  0.838  |  0.875  |  0.903  |  0.925  |
>    | Magnitude similarity |  0.878  |  0.908  |  0.933  |  0.950  |  0.962  |  0.971  |
>
>    Increasing the sampled batch size improves the similarity.
>
> 2. The final performance
>
>    We conducted experiments on Metamath-100k with LLaMA-2 7B with learning rate 5e-5 to assess the impact of batch size on final performance.
>
>    |     | 8 | 16 | 32 | 64 | 128 | 256 |
>    | :-: | :-: | :-: | :-: | :-: | :-: | :-: |
>    | MetaMath-100k | 52.79 | 52.99  | 52.91  | 53.56  |  52.57  |  53.22  |
>
>    Results indicate that larger batch sizes can lead to slightly better model performance, but generally stay around the same level. We recommend choosing a larger batch size (e.g., 64) to achieve relatively stable performance, if resources permit.
>
> ### Question 4
>
> **Question 4**: What would happen if the value of $r$ in $A$ and $B$ were swapped? Or if the index sets $I_{A}$ and $I_{B}$ were randomly chosen from $[1,2r]$?
>
> **Answer 4**: We test it using three schemes:
>
> - ArB2r: choose $I_{A} = [1, r]$ and $I_{B} = [r+1, 2r]$ (The scheme used in the paper).
> - A2rBr: choose $I_{A} = [r+1, 2r]$ and $I_{B} = [1, r]$.
> - Random: choose $I_{A}$ and $I_{B}$ in size r randomly sampled from $[1, 2r]$.
>
> |     | ArB2r | A2rBr | Random |
> | :-: | :--: | :-: | :-: |
> | MetaMathQA-100k | 52.79 | 52.38 | 52.01  |
>
> We found that the ArB2r scheme yielded the best results. Although equivalent in step 1, from step 2 onwards, the gradient of $B$ ($\nabla_{B}\mathcal{L} = (\nabla_W \mathcal{L}) A^T$) is larger than that of $A$ ($\nabla_{A}\mathcal{L} = B^T \nabla_W \mathcal{L}$). This can be seen as increasing the learning rate for matrix $B$. According to LoRA+[2], a larger learning rate for $B$ is beneficial, which might explain why ArB2r performs better.
>
> Thank you again for your insightful feedback. We will update our paper in the next version.
>
> [1]Amir, Idan, Tomer Koren, and Roi Livni. "SGD generalizes better than GD (and regularization doesn’t help)." Conference on Learning Theory. PMLR, 2021.
>
> [2]Hayou, Soufiane, Nikhil Ghosh, and Bin Yu. "Lora+: Efficient low rank adaptation of large models." arXiv preprint arXiv:2402.12354 (2024).

---

> > ### Comment · Reviewer_dSFq · 2024-08-09
> > **Thanks for your response.**
> >
> > Most of my concerns are addressed. So I'm willing to raise the score.

---

> > > ### Author Response · Authors · 2024-08-13
> > >
> > > Thank you for your response. We appreciate that most of your concerns have been addressed, and we would welcome any additional feedback. Please let us know if you have any further questions.

---

### Official Review · Reviewer_vjMb · 2024-07-12

**Soundness:** 3
**Presentation:** 3
**Contribution:** 3
**Rating:** 6
**Confidence:** 4

**Summary:**

LoRA has a slower convergence rate compared to full fine-tuning. This paper proposes a novel initialization method, LoRA-GA (Low-Rank Adaptation with Gradient Approximation), which aligns the gradients of the low-rank matrix product with those of full fine-tuning from the first step. Numerical experiments demonstrate that LoRA-GA achieves faster convergence and better or comparable performance to full fine-tuning, outperforming vanilla LoRA and its variants on several benchmark datasets.

**Strengths:**

-The paper introduces a novel initialization strategy for LoRA, enhancing its efficiency and performance without altering the architecture or training algorithm.
-The idea of aligning the gradients of the low-rank product with the full model’s gradients at the initial step is innovative.
-The combination of gradient approximation and stable scale for initialization is a unique contribution.

**Weaknesses:**

-The concept of using eigenvectors for initialization might not be entirely new, but its application in this specific context is original.
-The paper could benefit from a deeper exploration of potential edge cases or limitations of the proposed initialization method.
-Some sections, particularly those involving complex mathematical derivations, might be challenging for readers without a strong background in the area. Some of the detailed steps may need to be provided in the proof.

**Questions:**

See the weakness.

**Limitations:**

The authors have adequately addressed the limitations and potential negative societal impact of their work.

---

> ### Author Rebuttal · Authors · 2024-08-07
>
> We sincerely thank you for the valuable feedback and insightful comments.
>
> ### Question 1
>
> **Question1**: The concept of using eigenvectors for initialization might not be entirely new, but its application in this specific context is original.
>
> **Answer1**: Indeed, the idea of utilizing eigenvectors in initialization has been previously employed (e.g., PiSSA[1]). The novelty of our work lies in the approximation of the gradients in LoRA, with the eigenvectors being a result of this optimal low-rank approximation of the gradient.
>
> ### Question 2
>
> **Question2**: The paper could benefit from a deeper exploration of potential edge cases or limitations of the proposed initialization method.
>
> **Answer2**: Thank you for your suggestion. Although we did not encounter any edge cases in our experiments, we briefly discuss the possible edge cases below:
>
> 1. The matrix $A$ in our LoRA initialization is $O \left( \frac{d_{\mathrm{out}}^{1/4}}{d_{\mathrm{in}}^{1/2}} \right)$. Therefore, for certain extreme layers or MLPs where $d_{\mathrm{out}}$ is significantly larger than $d_{\mathrm{in}}$, the matrix $A$ in LoRA may experience numerical overflow. In our experiments, both Llama 2-7B and T5-base do not contain such matrices. We recommend that users of our method exercise caution by trying different stable scaling factor if their models include structures with such matrices to prevent potential numerical issues, or ensure careful management of numerical stability and precision.
> 2. The batch size used for sampling to calculate the gradient is crucial. Intuitively, when the batch size is only 1, we initialize LoRA using the gradient of a single sample. If this sample is an outlier, the initialization could degrade to the case of Gaussian initialization (random direction), or even worse, lead to a completely incorrect direction. We recommend using a larger batch size (at least 8) within the available computational power to avoid such edge cases and ensure better optimization.
>
> We will incorporate the above discussions in the next version of our paper.
>
> ### Question 3
>
> **Question3**: Some sections, particularly those involving complex mathematical derivations, might be challenging for readers without a strong background in the area. Some of the detailed steps may need to be provided in the proof.
>
> **Answer3**: We acknowledge that sections with complex mathematical derivations may be challenging for readers without a strong background in this area. We apologize for any possible confusion. In response, we will provide more details and explanations of our mathematical proofs in the revised version.
>
> Thank you again for your insightful feedback.
>
> [1] Meng, Fanxu, Zhaohui Wang, and Muhan Zhang. "Pissa: Principal singular values and singular vectors adaptation of large language models." arXiv preprint arXiv:2404.02948 (2024).

---

### Official Review · Reviewer_P55u · 2024-07-15

**Soundness:** 3
**Presentation:** 3
**Contribution:** 3
**Rating:** 6
**Confidence:** 3

**Summary:**

This paper proposes a novel initialization method for LoRA based on detailed theoretical analysis. The experimental results illustrate that the proposed method can achieve a great performance on the most tasks.

**Strengths:**

Strength:
1. The paper provide a beautiful theoretical analysis about the initialization method for the gradients of LoRA and full-parameter fine-tuning and then propose their method based on their analysis.
2. The proposed method is also very clear and easy to follow.
3. The provided results illustrate that the proposed method can achieve a better performance compared with the most LoRA variants.

**Weaknesses:**

Weakness:
1. I think the main weakness is from the experiments. I think the authors should provide more results on some complex tasks. For example, the paper mainly focus on GULU benchmark and metamath-100k. However, I think GULU has been solved by current PEFT methods. Then, for the metamath-100k, it selects 100k data from the vanilla metamath dataset. I recommend the author conduct their experiments on the full metamath dataset because the recent LoRA-based papers usually focus on the full dataset and therefore the readers can obtain a fair comparison and more directly to understand which method is better. I think metamath-100k is not a popular choice although Pissa also uses this dataset.
2. I am considering whether the authors tune the hyper-parameters of the baseline methods, because I find the results of some baselines are too weak. The best for LoRA-GA is usually not the best hyperparameters for other methods. I hope the paper can provide a fair comparison. That is also the reason why I suggest the author use a more popular dataset. For example, the default learning rate for the experiments on metamath is 2e−5, and the results on GSM8K are: LoRA (42.08), LoRA+(52.11), LoRA-GA(53.60). However, from the table 7 and table 8 in the appendix, we can find that the results of a larger learning rate 5e-5 in table 8: LoRA (46.89), LoRA+(55.23), LoRA-GA(52.79). That means increasing the learning rate from 2e-5 to 5e-5, the baseline LoRA can be improved from 42.08 to 46.89, LoRA+ can be increased from 52.11 to 55.23 and higher than LoRA-GA. So I considering whether we can obtain a better performance for the baselines when we further tune the hyperparameters. I think that is very important since the reader can better know whether each method can really improve the performance.


[1] LoRA Learns Less and Forgets Less.

[2] MoRA: High-Rank Updating for Parameter-Efficient Fine-Tuning.

[3] MiLoRA: Harnessing Minor Singular Components for Parameter-Efficient LLM Finetuning

**Questions:**

My question is about the hyperparameter selecetion for the baselines. Whether the hyperparameters the authors used in this paper is a good  selection.

---

> ### Author Rebuttal · Authors · 2024-08-07
>
> We appreciate your valuable suggestions and have conducted additional experiments in response to your feedback.
>
> ### Dataset Selection
>
> Regarding your concern about the complexity of tasks, we acknowledge that GULU may have been effectively addressed by current PEFT methods. Consequently, we expanded our experiments to the full metamath dataset to facilitate fair comparisons and align with recent LoRA-based research.
>
> ### Hyperparameter Tuning
>
> Initially, we tuned the hyperparameters using full fine-tuning and applied the same parameters to LoRA, LoRA+, and LoRA-GA. Based on your recommendation, we tuned the hyperparameters specifically for LoRA, LoRA+, and LoRA-GA on the full metamath dataset. This tuning was conducted by evaluating performance after training for one epoch, provided there were no numerical overflow or instability issues. We tested batch sizes of {32, 64, 128} and learning rates of {2e-5, 5e-5, 1e-4, 2e-4}. The optimal hyperparameters identified were:
>
> \- **Vanilla LoRA:** {128, 1e-4}
>
> \- **LoRA+:** {128, 5e-5}
>
> \- **LoRA-GA:** {128, 5e-5}
>
> The hyperparameter we obtained is similar to the result from hyperparameter search in [3]. For this experiment, we used greedy decoding instead of top-p sampling to better align with previous work.
>
> ### Results
>
> We observed that vanilla LoRA we implemented with rank 8 achieved comparable or superior results to those reported for LoRA with rank 16 in Figure S2 of [2]. The performance across epochs is as follows (average by 2 seeds):
>
> |                                   | Epoch 1   | Epoch 2   | Epoch 3   | Epoch 4   |
> | --------------------------------- | --------- | --------- | --------- | --------- |
> | LoRA (Rank=16, reported in [2])   | ~49       | ~54       | N/A       | ~58       |
> | LoRA (Rank=8, our implementation) | 55.19     | 58.37     | 59.28     | 58.90     |
> | LoRA+ (Rank=8)                    | 56.37     | **59.21** | 59.93     | 59.97     |
> | LoRA-GA (Rank=8)                  | **56.48** | 58.64     | **60.16** | **60.88** |
>
> These results demonstrate that LoRA-GA consistently outperforms both vanilla LoRA and, most of the time, LoRA+.  The initialization used in LoRA-GA contributes to improved performance across multiple epochs. Additionally, LoRA+ and LoRA-GA are orthogonal methods; combining them might yield even better results with careful design and we leave it as an interesting future direction.
>
> Thank you once again for your insightful feedback. We will incorporate these results and update our paper in the next version.
>
>
> [1] Yu, Longhui, et al. "Metamath: Bootstrap your own mathematical questions for large language models." arXiv preprint arXiv:2309.12284 (2023).
>
> [2] Biderman, Dan, et al. "Lora learns less and forgets less." arXiv preprint arXiv:2405.09673 (2024).
>
> [3] Pan, Rui, et al. "LISA: Layerwise Importance Sampling for Memory-Efficient Large Language Model Fine-Tuning." *arXiv preprint arXiv:2403.17919* (2024).

---

> > ### Comment · Reviewer_P55u · 2024-08-12
> > **Thanks for your response**
> >
> > Thanks for your response and new results.
> >
> > The authors have solved most of my concerns and I will raise my score to 6.

---

> > > ### Author Response · Authors · 2024-08-13
> > >
> > > Thank you for your response. We appreciate that most of your concerns have been addressed, and we are grateful for your willingness to raise the score. If you have any further feedback, please let us know!

---

### Decision · Program_Chairs · 2024-09-25

**Decision:**

Accept (poster)

**Comment:**

The paper introduces an innovative initialization method for LoRA that enhances convergence speed and performance of fine-tuning LLMs. The reviewers appreciated the novel approach, which aligns gradients of the low-rank adaptation with those of full fine-tuning at the first step, improving the convergence rate and model performance. Despite initial concerns about experimental complexity and hyperparameter tuning, the authors provided additional experiments and clarified their methodology in the rebuttal, addressing the reviewers' feedback. I recommend acceptance.